# ESPressoscope: A small and powerful approach for in situ microscopy

Ethan Li[3], Vittorio Saggiomo[2], Wei Ouyang[4], Manu Prakash[3], Benedict Diederich[1]*

**1** Leibniz-IPHT Jena, e.V., Jena, Germany, **2** Department of Agrotechnology and Food Sciences, Department of Bionanotechnology, Bornse Weilanden, University of Wageningen, Wageningen, NL, **3** Department of Bioengineering, Stanford University, Stanford, CA, United States of America, **4** Department of Applied Physics, Science for Life Laboratory, KTH Royal Institute of Technology, Stockholm, Sweden

* benedictdied@gmail.com

**Data Availability Statement:** All Hard- and Software-related files are available from in the github repository: https://github.com/Matchboxscope/Matchboxscope and further documented in the webpage:

## Abstract

Microscopy is essential for detecting, identifying, analyzing, and measuring small objects. Access to modern microscopy equipment is crucial for scientific research, especially in the biomedical and analytical sciences. However, the high cost of equipment, limited availability of parts, and challenges associated with transporting equipment often limit the accessibility and operational capabilities of these tools, particularly in field sites and other remote or resource-limited settings. Thus, there is a need for affordable and accessible alternatives to traditional microscopy systems. We address this challenge by investigating the feasibility of using a simple microcontroller board not only as a portable and field-ready digital microscope, but furthermore as a versatile platform which can easily be adapted to a variety of imaging applications. By adding a few external components, we demonstrate that a low-cost ESP32 camera board can be used to build an autonomous in situ platform for digital time-lapse imaging of cells. Our prototype of this approach, which we call ESPressoscope, can be adapted to applications ranging from monitoring incubator cell cultures in the lab to observing ecological phenomena in the sea, and it can be adapted for other techniques such as microfluidics or spectrophotometry. Our prototype of the ESPressoscope concept achieves a low power consumption and small size, which makes it ideal for field research in environments and applications where microscopy was previously infeasible. Its Wi-Fi connectivity enables integration with external image processing and storage systems, including on cloud platforms when internet access is available. Finally, we present several web browser-based tools to help users operate and manage our prototype's software. Our findings demonstrate the potential for low-cost, portable microscopy solutions to enable new and more accessible experiments for biological and analytical applications.

## Introduction

Microscopes are an essential tool in almost all areas of the life sciences. They are used to make analytical measurements, observe dynamic processes at cellular scales, and gain insights into small structures. The availability of consumer and hobbyist technologies like smartphones,

https://matchboxscope.github.io/ All datasets have been uploaded to Zenodo: https://zenodo.org/uploads/8211236 For long-term access, we have assigned the following DOI to the codebase: https://zenodo.org/doi/10.5281/zenodo.11179310.

**Funding:** This research was supported by a Grant from the German-Israeli Foundation for Scientific Research and Development (GIF, Grant number G-1566-413.13/2023).

**Competing interests:** NO authors have competing interests.

miniaturized image sensors, microcontroller development boards, single-board computers, 3D printers, and other rapid prototyping tools and platforms have revolutionized the microscopy field. These tools are approachable and low-cost, making it easier for more people to access and benefit from the latest advancements in microscopy. Innovative laboratories now operate commercial and custom-designed microscopes taking advantage of these technologies, enabling advanced imaging capabilities [1] for scientific exploration and discovery. For example, customized solutions have been developed for super-resolution imaging [2–5] and malaria diagnostics [6–8]; while modular platforms have been developed to simplify cutting-edge techniques [4, 9, 10]. These innovations increase the power of microscopy as an analytical tool [11]. A thorough review of handheld devices and imaging platforms for diagnostic, educational and professional use can be found in [12–14] or in our review [15].

Outside of traditional research facilities, the same advances in consumer technologies have enabled affordable, user-friendly microscopes that can easily be customized [16, 17] to meet specific experimental needs [18], and also to make microscopy accessible to a wider range of researchers and students. Many of these microscopes have been designed to promote STEAM education and research in previously underserved communities [19, 20]. An excellent example is the 2 dollar Foldscope [20] and its associated global community for citizen scientists to share what they see in the microcosmos. Other microscopes have been developed with the goal of making specific research techniques more accessible at high performance and low cost. One example is the 3D-printed and open-source OpenFlexure microscope, which is backed by a large user community (estimated 1k users based on number from the official forum http://openflexure.discourse.group, YouTube videos and manuscripts based on the OpenFlexure microscope), and is attempting be one of the first open-source medically-certified microscopes used to identify malaria in blood smears (https://openflexure.org/about/medical-devices). Another example is the Planktoscope [21], a low-cost autonomous stop-flow imaging microscope controlled by a Raspberry Pi to quantify the local biodiversity of plankton using a simple yet powerful fluidic system and microscopy components. This is backed by a very active user base inside a Slack channel (https://www.planktoscope.org/join) and an open-sourced commercialization of the device (www.fairscope.org). These examples show that global impact is achievable for high-performance devices which are made open and affordable, and which develop large user communities. This strategy aims to increase the use of microscopy in life sciences and beyond by providing accessible tools for education, medical diagnosis, and research.

Here, we aim to bridge the gap between the Foldscope and more expensive digital compound microscopes with a proof-of-concept for ESPressoscope, a pattern for designing low-cost ($\leq \$10 - 30$) and simple (build time of ca. 1–3 h) microscopy platforms based on consumer-grade microcontroller boards available worldwide. Other components are available off-the-shelf worldwide through online marketplaces such as AliExpress, Alibaba, Amazon, or Taobao. In addition to being compact, easy to source, and easy to build, our demonstration platform requires only a small number of additional hardware components for adaptation into the following five application-specific configurations described in this paper (Fig 1):

- "Matchboxscope": A portable brightfield microscope with a resolution between $3 - 4\mu m$ and a Wi-Fi interface. Multiple modifications are possible to improve illumination, stage movement, z-stacking, etc.

- "Anglerfish": An underwater microscope for studying aquatic microorganisms underwater

- "ESPlanktoscope": A simplified version of a flow-imaging microscope with an integrated peristaltic pump

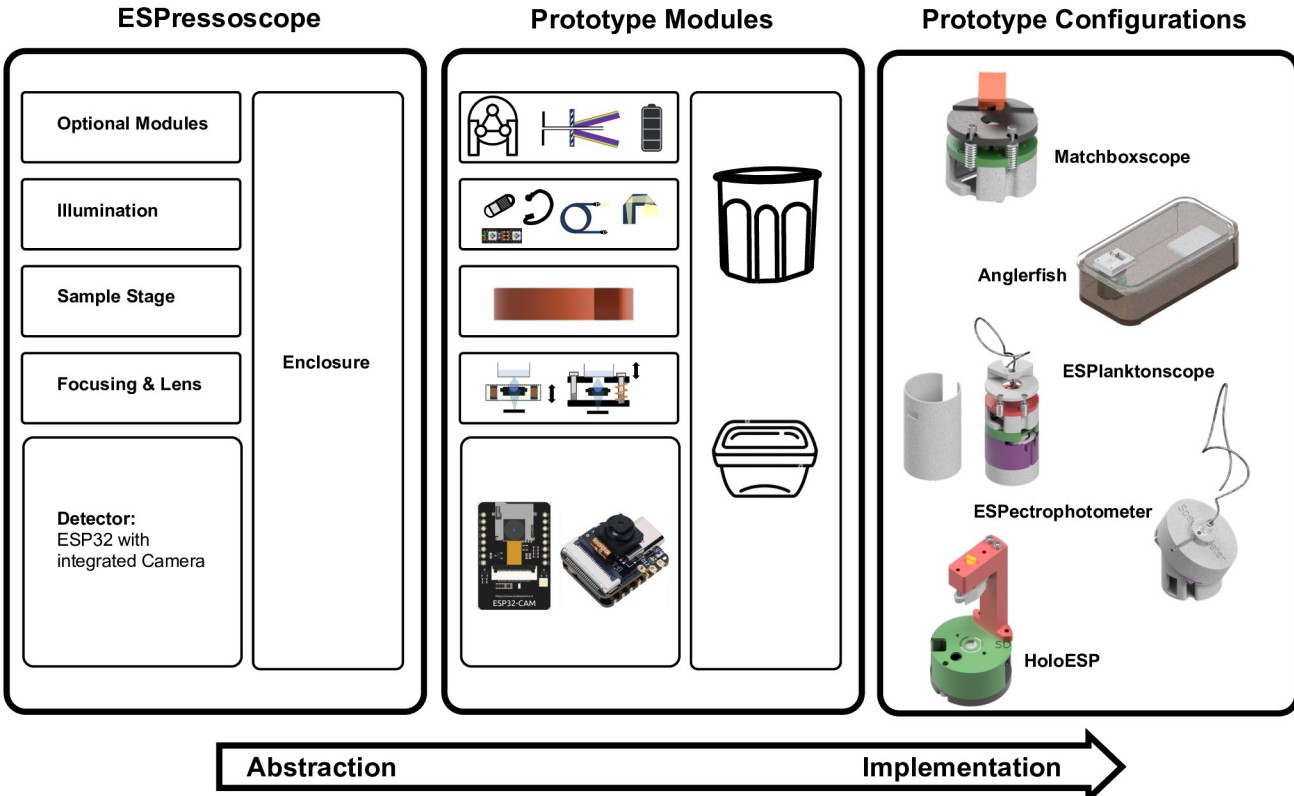

**Fig 1.** The ESPressoscope concept (left) consists of an ESP32 microcontroller development board with an embedded graphical user interface and an integrated camera which is combined with other modules which are selected depending on imaging configuration and which are combined through a layered structure. This paper demonstrates the concept with a prototype set of modules (center, refer to S1 Fig) which we combined in various ways to achieve a variety of prototype optical configurations, such as a compact general-purpose microscope (Matchboxscope), an underwater microscope (Anglerfish), a flow-imaging microscope with an embedded fluidic device (ESPlanktoscope), a spectrophotometer (ESPectrophotometer), and a lensless holographic microscope (HoloESP).

- "ESPectrophotometer": a compact, digital, low-cost visible-light spectrophotometer

- "HoloESP": A compact, low-cost, lensless holographic microscope

   In all configurations, the camera module board can act as a Wi-Fi access point for local and remote operation of microscope functionalities, either before deployment or throughout the entire deployment. A USB power bank or a lithium-polymer battery can power the board.

   An associated website (https://matchboxscope.github.io/) provides detailed documentation on how to set up and troubleshoot the prototype devices presented in this paper, and to enable users to flash the firmware for those devices directly from a web browser without any further installation of third-party software. This software infrastructure will help anyone to build and operate microscopes designed with the ESPressoscope approach.

   Our primary objective is demonstrate an approach for developing versatile, cost-effective microscopic imaging systems which various communities could independently adapt and use, and which would be suitable for in situ deployment in various conditions. To achieve this, we identified the functional requirements for the design of our prototypes to demonstrate the ESPressoscope concept, and then we defined the essential hardware requirements for these prototypes.

Our prototype devices should collectively be capable of imaging various kinds of samples, including samples attached to or mounted on microscope slides, as well as liquid samples with suspended particles, such as in microfluidic devices. Additionally, these devices should be deployable for extended durations in various remote settings with limited connectivity and power sources; extreme deployment scenarios may vary from inside incubators, such as for monitoring cell cultures, to underwater locations. This necessitates support for waterproof enclosures and unattended operation throughout several days and a mechanism to compensate for potential drifts in focus and ambient lighting throughout prolonged experiments. Finally, our prototype devices should be capable of high-contrast imaging. This requires support for oblique illumination, such as darkfield imaging. We aimed to meet these requirements while minimizing the cost and complexity of the devices by minimizing the overall number of parts required for their construction, and by developing a set of prototype modules which could be combined into configurations specialized for each use-case.

The core module of the ESPressoscope concept, shared across all prototype devices described in this paper, is an Arduino-compatible ESP32 microcontroller board with an integrated camera. A low-cost option is the ESP32-CAM board (Ai-Thinker, China, USD $5) in combination with its ESP32-CAM-MB programming board (China, USD $3). The ESP32-CAM board features an OV2640 CMOS Camera (Omnivision, USA, 1600 x 1200 pixels, 2.2 μm pixel size, Bayer pattern) comparable to image sensors in older smartphones and IP cameras. We have also implemented ESPressoscope device prototypes with a different ESP32 microcontroller board featuring the same camera sensor, the XIAO ESP32S3 Sense (Seeed Studio, China, USD $17). Although this board is more expensive, it is smaller and more reliable, and it has an external Wi-Fi antenna factor. Both boards have the option of mounting an SD card for file storage.

These ESP32 developer boards can be turned into a macroscopic imaging device simply by unscrewing their fixed-focus objective lenses (*f*#2.2, f' 4 mm). However, they can be adapted into high-magnification and high-resolution microscopes by adding a few off-the-shelf components (screws, light guide cables, optional electronics) and 3D-printed parts. In designing our prototype microscopes and the modules comprising them, we prioritized the supply-chain availability of necessary parts, the total cost of the resulting bill of materials, and the ease of independently replicating assembled devices. The resulting set of hardware modules enables a variety of configurations which may offer manual or automatic focus adjustment and transmitted light illumination.

## Results and discussion

### Matchboxscope: A simple microscope for small places

The simplest configuration, a Matchboxscope with a periscopic illumination mechanism (Fig 2a), can be replicated wherever an ESP32-CAM module and a 3D printer are accessible. Design files, all software, and an in-depth guide to replicating the device are available in the project's online repository at https://github.com/matchboxscope/matchboxscope. Further documentation is published at https://matchboxscope.github.io.

This initial configuration is a prototypical design suitable as a starting point for more specialized configurations. It can be used for everyday research tasks such as time-lapse imaging in incubators, imaging of microfluidic devices, and portable microscopy for fieldwork. It fits in a pocket (diameter: 52 mm, height: 20–30 mm) and can be powered by an external USB battery. The Matchboxscope's extreme portability and low cost make it suitable for citizen science projects. For example, one citizen science project sent Matchboxscopes together with simple

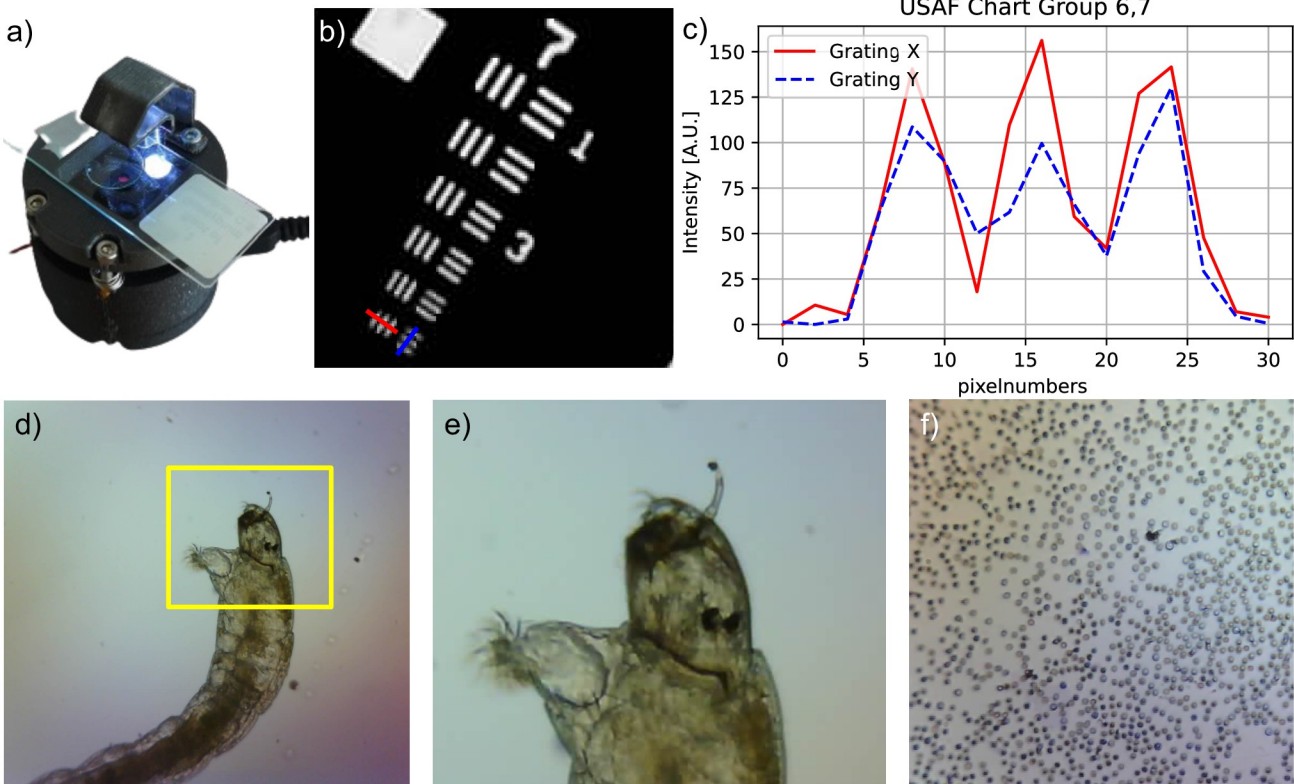

**Fig 2.** (a) A Matchboxscope configuration featuring periscopic illumination and spring-based focusing mechanism. (b) This simple microscope can resolve features as small as 4–5μm inside the USAF chart (group 6,7) also demonstrated with (c) the lineplot along the vertical and horizontal direction (red, blue). Examples of micrographs obtained with the Matchboxscope: (d, e) A mosquito larvae found in a pond and (f) red blood cells.

microfluidic chips [22] to participants for environmental monitoring of microplastics or plankton in water samples.

The simplest optical configuration for a Matchboxscope can be achieved by slightly unscrewing the integrated objective lens of the ESP32-CAM module. Unscrewing the lens moves the static focus from infinity to a finite distance closer to the camera. The magnification is given by $M = -a'/a$, $with d = a' - a$, with a the object distance and a' the image distance, and with a focal length defined by the camera lens $1/f' = 1/a' - 1/a$.

An image on the sensor is focused and magnified if $a < a$'. However, to create a compact device with the ESP32-CAM board's d 4 mm sensor and $f \approx 4\ mm$ focal length of the objective, we have chosen a' $\approx 18 mm$ and $\approx 4 mm$, leading to a total magnification of $\approx 4$ and an effective pixel size of $\approx 0.6 \mu m$. Then the ESP32-CAM board's f-number of f#2.2 allows a numerical aperture of $NA = 1/(2\cdot 2.2) \approx 0.23$ and thus an optical resolution of $d$ $(\lambda = 550 nm) = \lambda/(2*NA) \approx 1.2 \mu m$. Because we use the objective lens in a finite configuration, the effective aperture is smaller, and additional aberration (e.g. defocusing) may further degrade the imaging quality. This configuration maintains the Nyquist sampling criteria if the Bayer pattern is disregarded; otherwise, this configuration is considered to undersample slightly.

Since moving the camera lens would change the effective optical magnification, focusing would be simpler by moving the object relative to the lens to form a sharp image while keeping magnification constant.

## Focus adjustment

For this reason, we developed two possible focus adjustment mechanisms. For manual focus adjustment, the microscopy sample is placed on a 3D-printed spring-loaded or magnet-loaded sample holder which can be moved through the focus by turning one or more of three positioning screws located around the sample. For electronic autofocus, the camera objective lens' position is linearly adjusted with respect to a fixed camera sensor and sample; the position of the objective lens is actuated using a current-controlled voice coil actuator for M12 CCTV lens ("ML1608" 16X16X8 mm, China). The voice-coil actuator is controlled with pulse-width modulated (PWM) signal from the microcontroller, and it achieves a maximum linear motion of approximately 1 mm discretized with a resolution of 8 bits, leading to an axial step size of approximately $4\mu m$.

The electronic autofocus enables simple focus stacking, software-based autofocus to maximize the sample sharpness while varying the focus position, and extended depth-of-field (EDOF) measurements. However, it requires additional electronics and soldering, is vulnerable to slight variation in the magnification factor while focusing, and has a limited focusing range. S6c Fig shows a focus stack, where a permanent pen on glas was imaged while varying the lens' position. When electronic autofocus is used, we recommend using it for adjusting fine focus in combination with a manual focusing mechanism for adjusting coarse focus. The Matchboxscope firmware's contrast-based autofocus algorithm searches the Z position of the objective lens to maximize a relative sharpness metric. For a simple and fast metric, our implementation estimates relative image sharpness by comparing the number of bytes in the compressed JPEG image, since the JPEG compression uses discrete cosine transform (DCT) coefficients in such a way that input images with weaker components at high spatial frequencies will tend to result in an output with fewer bytes; this approach is also used by the Open-Flexure microscope for fast autofocus [23]. More sophisticated sharpness metrics, such as those calculated directly on the distribution of DCT coefficients [24], can be implemented if needed. S9 Fig gives an exemplary plot of the sharpness as a function of the lens position.

When images are in focus, the microscope achieves an optical resolution greater than 4 μm, as it resolves the smallest feature group in the 1951 USAF resolution test chart (Fig 2b and 2c).

## Illumination

The ESP32-CAM module's integrated LED can be used for sample illumination without additional electrical wiring. In the simplest variant of the Matchboxscope, light from this LED can be redirected to the sample with a light guide. This can be achieved with a simple 3D-printed periscope whose interior is coated with a reflective surface, such as a coating of waterproof ink from a silver pen marker (Edding 750, Germany), a spray coating of reflective paint, or a layer of reflective aluminum tape, to serve as a light source for transmitted-light microscopy. Alternatively, the periscope can be substituted with a TOSLINK fiber-optic cable normally sold for consumer-grade audio systems (Amazon Basics, 30cm, 4$). In both cases, the redirected light is guided from the ESP32 board to pass from the light guide's endpoint, through the sample, and into the objective lens; and the endpoint of the light guide defines the illumination mode, which can either be centered, oblique, or darkfield (i.e., outside the objective lens's NA).

For higher-contrast imaging in critical illumination, we found that an ultra-compact, watertight, coin cell-powered LED light, such as those used in balloon lights, produces a more homogeneous illumination of the field-of-view and is a suitable replacement for the ESP32-CAM module's integrated LED. Because this light is in a self-contained unit, it can be placed opposite from the objective lens on the other side of the imaging sample, removing the need for a periscopic or fiber-optic light guide. S4 Fig shows the stability of the light's intensity over

three days. The first trial of a Matchboxscope with this illumination configuration was to detect Schistosoma eggs using a size-selective microfluidic chip (see [22]).

Depending on experimental requirements, more sophisticated illumination methods can also be integrated with the Matchboxscope's hardware and software. For example, a USB-powered gooseneck lamp from IKEA (USD $4, Sweden), or a NeoPixel LED, LED ring, or LED array (Adafruit, NY, USA, 5–20$) can greatly improve the overall uniformity of illumination or add phase-contrast imaging by dark-field or oblique illumination and is integrated in the device's firmware. Alternatively, an LED-based system can be substituted with a laser pointer for fluorescence imaging (S6b Fig).

In all variants (see S8 Fig) of the Matchboxscope, the illuminating end can be moved freely and can be held using 3D-printed components to enable brightfield, oblique, and darkfield configurations. Example images obtained with these four different illumination mechanisms are shown for comparison in S3 Fig.

## Usage

To evaluate the Matchboxscope's ability to capture long-term timelapse series for in vitro live cell studies, we placed a Matchboxscope unit (spring-loaded manual stage, external LED light source, USB power bank) inside a cell incubator ($37°C$, 5% CO2, 100% humidity) to capture images every minute over three days to track the growth of a dish of HeLa cells (S1 Video, S4 Fig). The microscope exhibited imaging drift (200 μm, mostly in the x/y direction) during the first 30 minutes of the timelapse. Thus, the microscope's mechanical structure requires time to warm up to stabilize before timelapse imaging. At the end of the timelapse, we measured a total drift of about 100 μm within the x/y plane. We hypothesize that this drift is caused by thermal expansion and softening of the Matchboxscope's 3D-printed thermoplastics, and that the amount of focus drift heavily depends on parameters like materials used, 3D-printing strategy (e.g. infill pattern), and environmental conditions around the Matchboxscope. S4 Fig shows the lateral drift during a time lapse at room temperature ($20C$, 40% humidity), which is significantly lower and lies at around $\pm 4\mu m$.

Even though the Matchboxscope's design features an air gap between the sample and the ESP32 camera module, heat buildup in the electronics caused a slight increase in the temperature of the sample-containing petri dish during the cell incubator timelapse experiment. This led to the formation of condensation droplets on the lid which caused non-uniform illumination. This excess heat also induced apoptosis for cells in the dish (S5 Fig); this phenomenon was not observed in an experimental control placed next to the microscope. For this experiment, we had set the Matchboxscope to operate in "continuous" mode (Â $150mA$ current draw while capturing/streaming images) with an internet connection allowing us to remotely access the microscope's control interface to monitor the progress of the experiment. The Matchboxscope's "autonomous" deep sleep mode would be more appropriate, due to its lower power consumption (Â $20\mu A$ current draw) and thus lower generation and transfer of heat to the imaging sample between timelapse events. Alternatively, we could have installed a fan to generate airflow between the imaging sample and the Matchboxscope's camera in order to mitigate heating of the sample from the Matchboxscope.

The Matchboxscope with a magnet-loaded static focus and periscopic illumination configuration was also preliminarily characterized and compared to other microscopes for detecting Schistosoma parasite eggs inside a microfluidic device in [22]. Schistosomatosis is a neglected tropical disease which affects nearly 140 million people annually [25]. Thus, the ESPressoscope concept can be implemented as a low-cost, portable imaging device for deployment in remote areas to detect Schistosoma parasite eggs in urine samples.

## Anglerfish: A submersible microscope for underwater time-lapse imaging

The study of plankton, biofilms, and other biological dynamics in aquatic surfaces is highly relevant to microbiology, and an in situ imaging tool for studying such dynamics could provide further insights into understanding them. For example, the composition of microbial communities in biofilms can influence processes such as biofouling and even the ecological fate of microplastics [26, 27]. While many researchers have successfully studied biofilm growth under in vitro laboratory conditions, studying their dynamics directly in water has been difficult. Watertight or underwater microscopes in holographic [28], fluorescence [29], or brightfield [30] configurations mostly measure the composition and behaviors of planktonic organisms in fixed volumes rather than measuring microbial colonization processes on submerged surfaces. Furthermore, such microscopes are usually custom-designed, complex, and expensive, with costs as high as—and exceeding—USD $100k.

Here, motivated by open-source and frugal science technologies, we opted for a radically different approach to design Anglerfish, a timelapse-imaging microscope appropriate for recording biofilms, microalgae, plankton, and other aquatic microorganisms in situ, which encloses an ESP32 camera module in a water-tight enclosure. Initially, we upcycled glass fruit preserve jars as water-tight enclosures for the Matchboxscope's electronics and other water-sensitive components. An alternative approach is to use watertight plastic food containers (e.g. Emsa Clip-and-Close, Germany, USD $4; or Ikea watertight lunchbox "Flottig", USD $10), which have a less constraining form factor and also can be integrated with a removable microscope slide where biofilm formation is observed (Fig 1). In this case, the microscope slide can be retrieved after the end of in situ time-lapse imaging for post-experiment microscopy with lab instruments at higher magnification and using more sophisticated techniques (e.g. fluorescence microscopy).

**Focus adjustment.** Our Anglerfish prototype features a voice coil-actuated electronic focusing mechanism so that the microscope operator can determine and adjust the focus of the microscope after submerging it. The microscope performs focus bracketing by saving a stack of images while sweeping the focus distance around the desired focus setting at every timelapse event, enabling post-experiment compensation for any small focus drift throughout long timelapse imaging series or, with more complicated image processing, focus stacking.

**Illumination.** For sample illumination with Anglerfish, we recommend using a self-contained waterproof battery-driven LED light which outputs light at a high intensity, as it can achieve high imaging contrast despite biofilm growth and requires no additional wiring. This external light module could be substituted with a combination of the ESP32-CAM's integrated LED and a periscope or TOSLINK cable light guide, but such a substitution will produce weaker illumination and lower imaging contrast. In experiments, we found that the self-contained external LED light can be left on continuously for multiple weeks with one set of batteries. To compensate for the loss of illumination intensity as the battery discharges throughout a deployment, the Anglerfish's ESP32 firmware uses exposure bracketing to acquire and store images on the SD card with a range of exposure parameters for each layer of the focus bracket, enabling selection of images with the best illumination conditions after microscope recovery. This bracketing approach removes the need to rely on automatic white balance, exposure control, and gain control algorithms which could choose inappropriate parameter values in autonomous deployments. This approach enables in situ imaging of diverse biofilm samples without requiring extensive pre-experiment tuning of imaging parameters.

**Power.** To maximize power efficiency, the Anglerfish's ESP32 camera module remains in a low-current deep-sleep mode between timelapse events. Electronic components are powered by a rechargeable lithium-ion battery inside the Anglerfish's water-tight enclosure. We found

that some USB power banks shut down when the current draw falls below some threshold for a given time; such power banks will not power our ESP32 microcontroller boards properly in "deep sleep" mode, because the ESP32 camera module's power consumption is too low during deep sleep mode and thus will cause automatic shutdown of the power bank. This problem can be avoided with a DIY power bank in which a lithium-ion battery connected to a 5 V DC step-up converter constantly supplies the ESP32-CAM module with power. The ESP32-CAM can be substituted with a Seeed Studio XIAO camera board, which includes an integrated battery driver so that the lithium-ion battery can instead be connected directly to the board. On average, the ESP32 microcontroller in either board consumes approximately 1000$mAh$ of power per 24-hour period when configured to perform an exposure bracketing series (1/2/5/10/20/50/100/200/500 ms) every minute.

**Improvement with UC2 modules.** After setting the focus and initiating time-lapse imaging, the resulting assembly remained operational underwater for at least five days (10,000 mAh power bank, 1 image/minute). The device was functional, but we could not properly capture biofilm formation due to image focusing challenges, even with our focus-bracketing mechanism. Setting up a clear image using the voice coil motor was challenging due to a change in focus upon immersion in water, as water would fill a gap between the lunchbox lid and the glass slide. Further defocusing occurred when transporting the microscope to the underwater deployment site, since small mechanical shocks and vibrations in transit also altered the position of the microscope with respect to the glass surface, resulting in the image being defocused beyond the limits of the voice coil motor's focusing range. While refocusing was still possible with manual focus adjustment immediately before deployment, the Anglerfish design did not lead to reproducible results in its original implementation, and a larger electronic focusing range was needed.

For these reasons, we modified this version by combining the ESPressoscope concept with modules from the UC2 toolkit of open-source microscopy modules [3]. We formed a finite corrected microscope by combining a Seeed Studio XIAO Sense ESP32 camera board with a customized UC2 cube insert and a motorized z-stage (50mm travel range, NEMA11 stepper motor) holding a 10x 0.25NA finite-corrected objective lens. This assembly fit in the same waterproof container and was successfully deployed in a preliminary test with a 24-hour time-lapse in the Leutra river in Jena, Germany (Fig 3c). In this test, the microscope was operated without the power-saving deep-sleep mode, resulting in high power consumption from the Wi-Fi electronics throughout the experiment.

Using the Anglerfish's web browser-based GUI, we achieved precise focus of the objective relative to the Anglerfish's mounted glass slide so that its water-facing surface was in focus. The larger focusing range with the motorized z-stage simplified the focusing of the sample, and the higher-quality objective lens significantly improved image quality and ease-of-use in our test. The Anglerfish's autofocusing algorithm successfully compensated for temperature-induced focus drift resulting from the day-night cycle.

The 24-hour timelapse was acquired with flat-field correction of the image sensor performed in Fiji after downloading all images to a local computer [31] in order correct for illumination inhomogeneities. During the experiment, several organisms were observed crossing the field of view (Fig 3d and 3e). Especially after 20 h, many small organisms were recorded on the glass surface (S2 Video). However, the microscope's resolution was insufficient to identify smaller details. One reason for this is that the lunchbox contributes to optical aberration and scattering, which reduces the maximum possible resolution. One solution would be to drill a hole in the lunchbox and seal it with a glass slide, at the risk of compromising watertightness. An alternative solution could be to replace the lunchbox with a more expensive IP66 electric enclosure with a clear window.

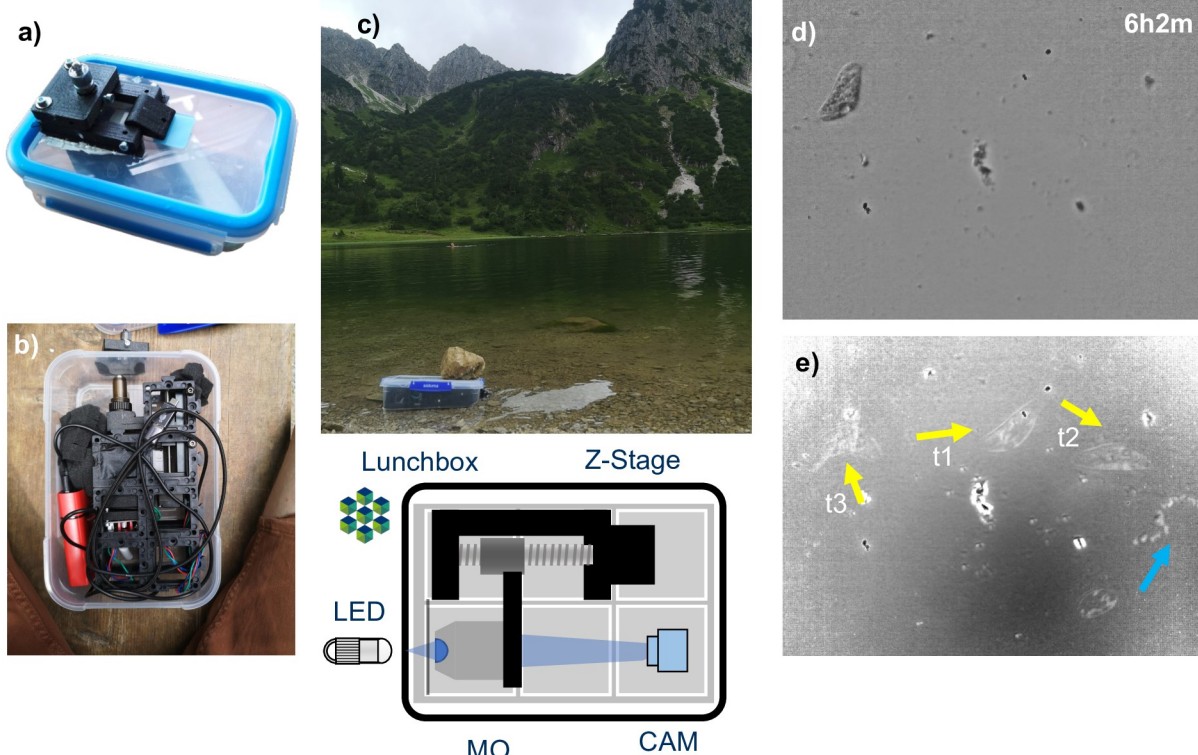

**Fig 3.** The Anglerfish: a submersible and automatic microscope for underwater operation, built using (a) a Matchboxscope with additional 3D-printed components, and (b) by combining an ESP32 camera module together with modules from the UC2 microscopy toolkit. With deployment in a pond (c) and application of flat-field correction to acquired images, the microscope allows observation of initial biofilm formation and microbial behaviors, including (d) a Paramecium crossing the field-of-view. (e) A variance projection of the timelapse visualizes the movement of different particles over time, with yellow arrows indicating the movement of the Paramecium at different timepoints and the blue arrow indicating the movement of a smaller microorganism. The optical path is visualized as a ray diagram, which also indicates the resulting magnification (*a* represents the distance from object to the lens, *a'* the distance between the lens and the camera, *d* the overall distance between sensor and sample $d = -a + a'$).

Our results with the Anglerfish configuration demonstrate the ability for ESPressoscope configurations to be upgraded for improved imaging performance depending on operational requirements, including through combination with modules from other microscopy hardware toolkits.

## ESPlanktoscope

Thanks to the layered hardware architecture of the ESPressoscope concept, additional modules can easily be stacked on top of each other to integrate additional functions. To this end, we prototyped an ESPlanktoscope configuration (Fig 4) by adding a small 3D-printed peristaltic pump derived from an open-source hardware design [32]. Like in the original Planktoscope concept [21], the flow rate of a liquid sample across the ESPlanktoscope's field-of-view can be controlled to image individual microscopic particles suspended in a large volume of input sample.

The ESPlanktoscope's 3D-printed peristaltic pump enables the microscope to continuously measure samples at a high rate (approx. 500 µL/min). To this end, the flow can be controlled in positive and negative directions to move liquid back and forth. Ball bearings reduce the frictional resistance between the rotor and tubing during the pumping process so that the torque

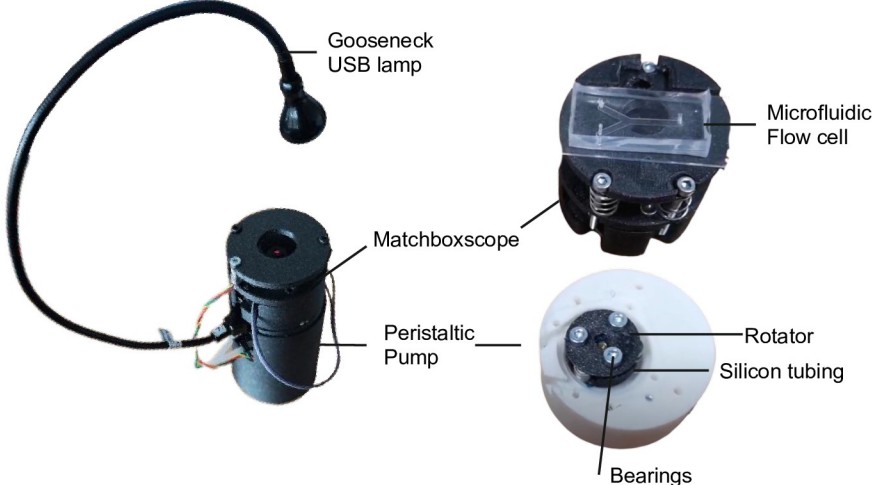

**Fig 4.** The ESPlanktoscope combines the standard Matchboxscope (top) with an additional ESP32 board for driving a peristaltic pump. Being a Matchboxscope extension, all possible Matchboxscope modifications explained before (different light sources and focusing capabilities) can also be applied to the ESPlanktoscope.

provided by the pump motor is sufficient to pump larger samples at a flow rate through the field of view of the microfluidic flow cell.

The ESPlanktoscope's peristaltic pump is actuated with a stepper motor (28byj-48, 5$, China) which is controlled by a ULN2003 (Texas Instruments, Texas, USA) motor driver. For an ESPlanktoscope using the ESP32-CAM module, which does not provide enough GPIO pins to control the motor directly, the motor driver must be controlled by a second ESP32 micro-controller board connected to the ESP32 camera module, as the ESP32 camera module does not offer enough GPIO pins to control the motor directly; in such a configuration, the ESP32 camera module sends a PWM signal to the second microcontroller in order to set the flow rate of the pump. By contrast, the Seeed Studio XIAO camera board provides enough GPIO pins to control the motor driver directly without the requirement for a second microcontroller board.

Fig 4 shows an assembled ESPlanktoscope unit equipped with an IKEA lamp which can be used for brightfield, oblique, and darkfield illumination. This gooseneck lamp provided a high degree of flexibility to align the light source relative to the homemade microfluidic chip and relative to the tubing connecting the microfluidic chip to the pump. A video showing the device in operation can be found in S3 Video.

Our prototype ESPlanktoscope configuration demonstrates the flexibility of the ESPresso-scope concept for integrating additional hardware modules to provide new imaging capabilities.

### ESPectrophotometer

Many different open-source designs exist for making compact digital spectrophotometers by combining a camera with a reflection grating (e.g. a CD/DVD) or a transmission grating [33]. This inspired us to use the ESPressoscope approach to design a digital spectrophotometer, the ESPectrophotometer.

Our ESPectrophotometer prototype is implemented by adapting a low-cost transmission grating (Diffraction Grating Slide-Linear 1000 Lines/mm, 1$) with a tilt angle of 35° in front of the camera. The entrance slit for light consists of an FDM 3D printer hotend nozzle

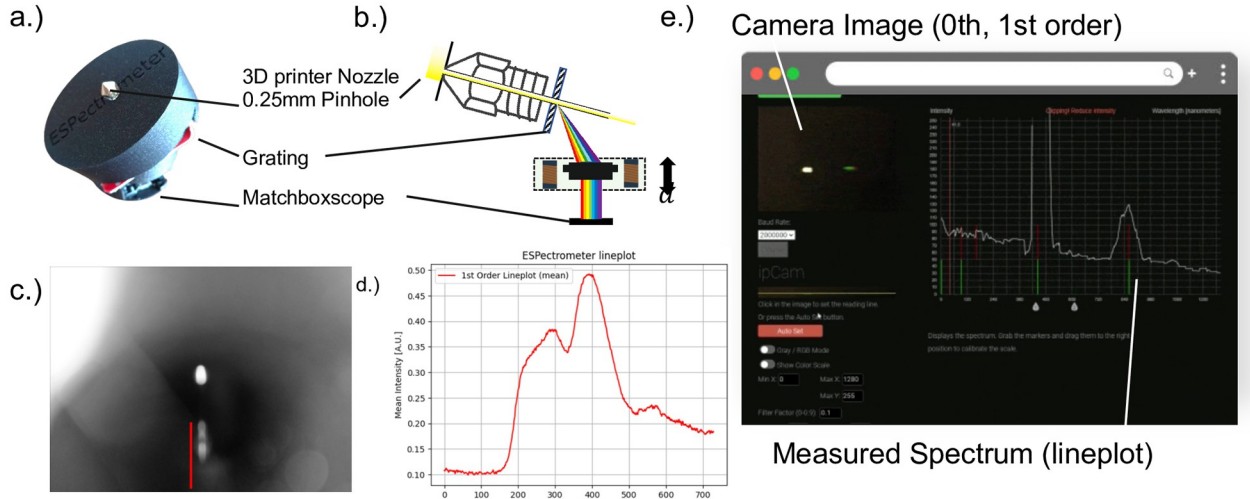

**Fig 5.** (a) Adapting the Matchboxscope into a spectrophotometer the objective lens to be mounted on the camera and a 3D printing nozzle to be added as a pinhole. (b) An adjacent grating diffracts the light before it gets focused on the sensor. (c) The resulting image on the camera sensor can be converted into a spectrum by drawing a d.) line plot through the intensity signal along the first diffraction order. e) A browser-based tool derived from the [34] receives frames from the ESP32 camera and draws the lineplot. For quantitative pixel/wavelength measurements, the ESPectrophotometer would require calibration.

(0.1 mm, eBay, 5$) placed approximately 40 mm from the grating. This off-the-shelf brass-machined part guarantees a much higher precision compared to 3D-printed designs. The 3-D printer nozzle creates an angular independent point source of light which passes through the tilted diffraction grating. Only the first diffraction order enters the camera module, which is equipped with an objective lens to focus the spectrum on the sensor.

Software running in a web browser receives camera frames from the ESP32 camera module using the WebSerial browser API, while open-source client-side Javascript software adapted from [34] directly processes and visualizes the spectrophotometry data. In this case, the ESP32 camera module is connected to the computer using a USB cable rather than Wi-Fi. The software is provided as part of ESPressoscope's freely-accessible documentation bundle (https://matchboxscope.github.io/spectrophotometer/espectrophotometer.html) and does not require installation. The software allows the user to select the position of the spectral components before a 1-D spectrum is displayed as a line plot. In Fig 5c–5e, we measured the spectrum of a cellphone's flashlight. The spectrum exhibits a clear dip in the green region of the white light spectrum as is typical for white LEDs. We have not characterized spectral resolution and accuracy in our preliminary demonstration of the feasibility of the ESPectrophotometer as a realizable adaptation of the ESPressoscope concept.

Our prototype ESPectrophotometer configuration demonstrates the feasibility of developing ESPressoscope devices which integrate real-time image processing and visualization, as well as the versatility of the ESPressoscope concept for applications beyond the life sciences.

## HoloESP

Lensless imaging provides the ability to capture in 2.5D and thus refocus an acquired image with digital post-processing. By contrast, the Matchboxscope's fixed-focus setting poses challenges for dynamic imaging. This limitation can be solved by inline holographic imaging as implemented by the HoloESP configuration of the ESPressoscope platform. A simple narrow-

band LED (450nm +/-20nm) is mounted approximately 80 mm from the sample and spatially filtered with an FDM 3D printer hot-end nozzle (0.1 mm). This creates a point source of light which illuminates the sample with weakly coherent spherical waves, producing a very simple and inexpensive inline holographic microscope. The captured holograms are numerically backpropagated using a custom PyScript module which can run in the web browser of a computer connected to the HoloESP microscope. This algorithm numerically refocuses the raw hologram and outputs a real-time display of its back-propagated result. To image small particles such as dust particles, we removed the black plastic cap holding the threaded objective lens provided with the ESP32-CAM module's camera sensor and instead applied the sample droplet directly to a coverslip placed directly on the sensor window. According to the Nyquist criterion, the approximate resolution should be about twice the pixel size (3 μm). Fig 6 shows the basic working principle, and some numerically refocused microplastic particles that were found in shower gel; the backpropagation algorithm successfully refocused the raw image into the sample plane and formed a sharp image of the microparticles (Fig 6c–6e). The HoloESP achieves a resolution of approximately 3 μm measured by a line plot at the smallest visible structures.

Imaging of E. coli bacteria suspended in a liquid sample produced images with very little contrast, resulting in a holographic reconstruction with a very low signal-to-noise ratio. Over time, E. coli sunk down and started attaching to the glass surface; this process was captured by the HoloESP's camera. However, due to the low signal-to-noise-ratio (SNR) of acquired data, it was impossible to reconstruct the holograms to identify individual bacteria.

These results with our prototype HoloESP configuration demonstrate the preliminary feasibility of developing ESPressoscope devices which integrate computationally expensive real-time image processing algorithms.

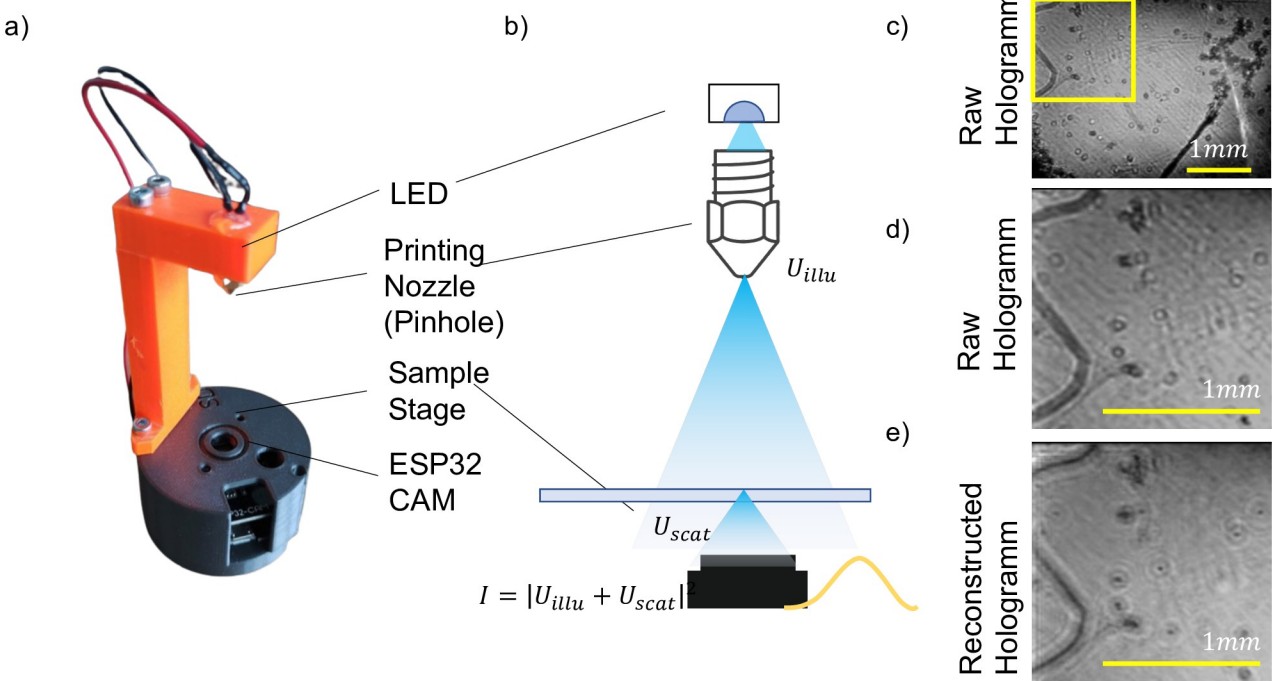

**Fig 6.** (a) Inline holographic microscope, with an orange arm holding an LED whose output is spatially filtered by an FDM 3D printer's hotend nozzle. (b) The point source of light sends out spherical waves so that scattered and unscattered waves interfere on the sensor. Using the Fresnel transform, the raw digital hologram (c, d) can be numerically refocused as in (e).

## Firmware and user interface

An easy-to-use digital microscope should be operable with a user interface which is simple and intuitive and which doesn't require complex installation procedures. Basic functionalities such as focusing, image acquisition, timelapse initiation, and configuration of camera and imaging parameters should be easily accessible through this interface, and this interface should be easy to operate with minimal software and hardware tooling. ESP32 microcontroller boards are compatible with the Arduino Integrated Development Environment (IDE) which offers a wide range of libraries useful for building firmware with a user interface that can be operated from the web browsers of devices with Wi-Fi connectivity, such as computers and smartphones. The use of a web browser-based user interface offers several advantages. First, it eliminates the need to install software on devices external to the microscope, making it more accessible to a wider range of users. Second, it allows for easy access to the microscope's functionality from any device, regardless of its operating system or configuration.

In implementing the firmware for the various ESPressoscope configurations, we used available third-party libraries to achieve a fully functional prototype. Additionally, in order to avoid requiring users to install third-party software for compiling or flashing our firmware onto the ESP32 module, we used the experimental Web Serial API as part of a web page (https://matchboxscope.github.io/firmware/FLASH.html) which can communicate with an ESP32 camera module connected by a USB cable; this allows the web page to upload firmware to the module, but currently only recent versions of Chromium-based web browsers implement the Web Serial API. Despite the limited browser support, this zero-cost, zero-installation setup process is important for reducing the expertise required to bring up, operate, and maintain a scientific instrument meant to be widely accessible. The firmware binaries are automatically compiled on cloud infrastructure with a continuous integration pipeline running in Github Actions (S6 Fig). Firmware binaries can be uploaded using either an over-the-air (OTA) mechanism or through the Web Serial-based web page, which provides a selection of customized firmware configurations (e.g. for the ESP32-CAM board, the Seeed Studio Xiao Sense board, Serial Camera, ESPectrophotometer).

The resulting software has the following functionalities, all of which are accessible through the browser-based GUI (S7 Fig):

- Wi-Fi network and internet connectivity management

- Live preview from the camera for exploring samples and adjusting focus

- Capture and saving of images to the ESP32 camera board's attached micro-SD card

- Timelapse imaging, where the microcontroller periodically captures images, saves them to an external micro-SD card, and otherwise stays in a deep sleep mode to reduce power consumption (S2 Fig)

- Focus adjustment, where the lens can be moved to increase focusing contrast

- Configuration of camera and image acquisition parameters, such as exposure time, gain, resolution, and timelapse interval

- Control of hardware parameters specific to optional hardware modules, for microscopes with those modules

- Image processing, using the built-in integration with ImJoy and ImageJ.js [35] Uploading of new firmware using the Over-the-Air (OTA) update feature

All functionalities are implemented using an HTTP-based REST API and a Javascript-based frontend which is served by a web server running on the ESP32 camera board's micro-controller. Alternatively, we provide a very basic Android app (https://matchboxscope.github.io/docs/APP) which uses the REST API and has the ability to save images from the microscope to the phone's storage. This is especially helpful for acquiring live video clips from the microscope, as higher image acquisition can be achieved by downloading frames via HTTP and saving them from a computer than by having ESP32 camera module to save individual frames on its attached micro-SD card.

## Conclusions and outlook

In this work, we aimed to prototype and evaluate a cost-effective approach to developing microscopy systems for many different settings and use-cases, including in situ deployments in isolated or otherwise challenging environments. We have demonstrated the ESPressoscope concept as a design pattern which enables people to make mission-specific trade-offs between various priorities—component availability, component cost, simplicity of assembly, provided functionalities, and imaging quality—by specifying a variety of recomposable modules which can be substituted for each other in the design of a specific microscope configuration. We have also introduced and preliminarily characterized some specific ESPressoscope device proto-types as easy-to-replicate designs which enable a variety of deployment scenarios. In particular, the Anglerfish device is—to the best of our knowledge—the first low-cost underwater micro-scope which aims to be reproducible by a layperson and to be deployable in situ at larger scales. We believe that the kinds of observations Anglerfish enables for microbiota on aquatic surfaces in ecological contexts may provide new information about the biodiversity and behaviors of aquatic microbial communities.

The presented prototypes of our ESPressoscope concept have a number of limitations and problems resulting from our choice of specific hardware components. For example, the ESP32-CAM module has been reported by some users to behave unexpectedly and to break for no obvious reason. This may be due to manufacturing mistakes, inappropriate selection of components such as the power controller, or design issues with the ESP32-CAM module itself. One known design issue of the ESP32-CAM board, for example, is inadequate isolation of the ground planes, leading to occasional noise on the board's input pins. This lack of reliability may lead to unexpected errors in the firmware of devices or even in the failure of a timeseries experiment. Because the ESP32-CAM board is an interchangeable module within the ESPressoscope approach, any problems inherent to the board can be solved by substituting it with an alternative ESP32 camera board—potentially increasing the total cost of the microscope's parts—as long as the substitute board can be integrated into the ESPressoscope's mechanical structure with a 3D-printable adapter. We have already done this with the Seeed Studio Xiao Sense board for the Anglerfish configuration with UC2 modules. Similarly, the ESPressoscope's modularity architecture enables other iterative improvements to be made in the design of our described modules and configurations—as well as new modules and configurations—to improve any aspect of system performance.

The prototype modules and imaging devices described in this paper all emphasize low cost and ease of replication over image quality in their design tradeoffs, and thus they achieve lower image quality compared to other open-source systems such as the OpenFlexure micro-scope [8]. Nevertheless, these devices may still be useful for many microscopy-related projects in field research and in educational settings (exemplary shown in Fig 7), whether on their own or in combination with other microscopy systems. With the provided capabilities, low cost, and use of widely-available parts in the specific designs we have described, we

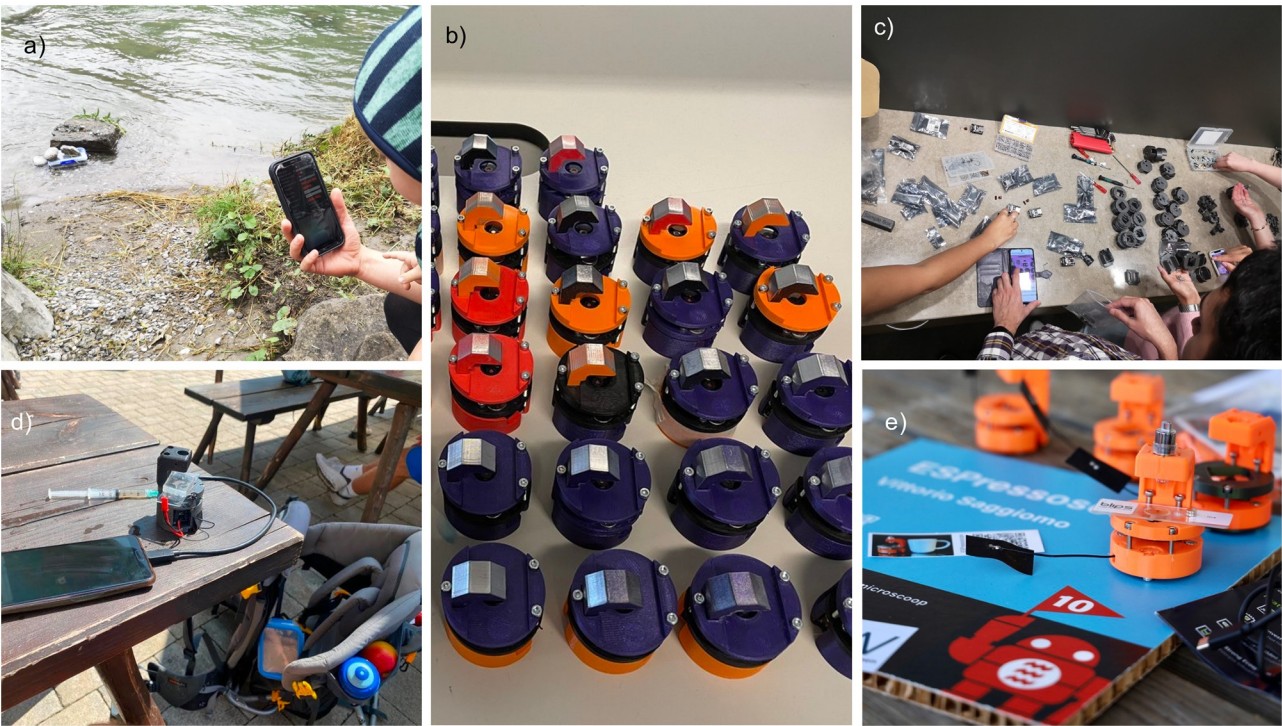

**Fig 7.** a) The light weight and small size of the minimal Matchboxscope configuration make it suitable in remote locations for on-site measurements, as already demonstrated by various users under the hashtag "#matchboxscope" on Twitter/X. (b) The Matchboxscope can easily be mass-produced to deploy it in different settings such as field work (d) or STEM education (c). (e) The recent Seeed Studio XIAO Sense ESP32 camera board makes the device even more compact, as presented at the 2023 Eindhoven Maker Faire.

believe these microscopes could become valuable image data collection tools for many communities.

The design process of our ESPressoscope device prototypes followed a tight feedback loop with users whose task was to replicate the design only with the help of our provided online documentation during a series of workshops in multiple countries (e.g. Nigeria, Netherlands, Germany, USA UK). In these workshops, we noticed that different users may engage with documentation in different ways depending on their contexts and prior experiences, leading to variability in the assembly process and in the image quality of the resulting devices. Participant feedback from these workshops also helped us to improve the online documentation for users beyond these workshops. Ahead of this manuscript, users have already started posting on social media under the hashtag "#matchboxscope" about new microscopes being set up and new data being collected, which gives us an idea of who is using this microscope and why.

User feedback on social media as well as on GitHub and in a dedicated non-representative survey has improved our prototype devices during the still-ongoing open development process. It also demonstrates that pre-publication and active interaction with the community can facilitate improvements to projects, as users of our prototypes were involved early in the design process and experimentation. We hope our development of the ESPressoscope approach contributes to research on how Open Science can be designed and conducted with the help of open instrumentation. Now, we invite the community to improve upon our described designs by developing new modules and devices using the ESPressoscope concept.

## Supporting information

**S1 Fig. Exploded view of the matchbox microscope.** The unscrewed photo objective lens serves as the microscope imaging lens. An in-focus image is formed when $\frac{1}{f'} = \frac{1}{a} - \frac{1}{a'}$. The various elements of the device, such as the stage or the illumination, can be exchanged and combined depending on the specific imaging application.
(PDF)

**S2 Fig. Operation of the firmware.** The flowchart shows the operation of the firmware, where the "Anglerfish-mode" is for autonomous operation, allowing the device to take images periodically and go into the ESP32 microcontroller's deep sleep mode to reduce energy consumption. Alternatively, the user can control the microscope from the browser, observe its camera stream, and perform image processing tasks.
(PDF)

**S3 Fig. Variation of imaging contrast for various illumination mechanisms.** A prepared slide from a Gossypium stem is used to illustrate how each illumination mechanism affects the illuminated field of view, visible contrast, homogeneity, and signal quality. Different mechanisms provide varying degrees of freedom, contrast, and usability, such as periscopic illumination, battery-driven LED, Toslink fiber-optic cable, Ikea USB lamp, and Neopixel LED matrix.
(PDF)

**S4 Fig. Intensity variation over time with different exposure times.** The graph shows the average intensity of images taken every minute and stored on the SD card with different exposure times (1 ms, 5 ms, 10 ms, 50 ms, 100 ms, and 500 ms). The microscope was placed in a dark box with the external battery-powered LED light switched on. Even after 3 days of constant illumination, the image intensity at longer exposure times is sufficient for brightfield microscopy. Lateral drift of the time series, calculated as a cross-correlation of successive images, shows a sudden jump in value after about 8 hours, likely due to the opening of a door.
(PDF)

**S5 Fig. Timelapse series of HeLa cells.** HeLa cells in a standard 25mm petri dish are shown. After around 2 days, the cells stopped replicating, most likely due to the increased temperature of the ESP32 camera module.
(PDF)

**S6 Fig. Example images from the Matchboxscope.** (a) Samples from a nearby pond. (b) Fluorescence capture using a blue laser pointer (450nm, laserlands.net) and a gel colour filter (LEE #13) on a yellow pen marker (Stabilo, Germany) on paper. (c) Focus stack of a line drawn by a marker on a coverglass on the surface of the Anglerfish microscope, showing cross-sections along z in x and y.
(PDF)

**S7 Fig. Continuous integration pipeline on GitHub Actions.** The pipeline is used to compile the firmware binaries, build the documentation website based on the open-source documentation building system Docusaurus, and host the image gallery of images uploaded from the ESPressoscope firmware. The firmware on the ESP32-CAM module runs in either interactive mode through a web browser or fully autonomous for timelapse imaging.
(PDF)

**S8 Fig. Different instrumentation configurations.** Various configurations are created by combining an ESP32 camera module with additional components to realize specific functionalities. The ESPressoscope architecture is completely modular, allowing different add-ons to

be used interchangeably with various configurations. Options listed in parentheses "(X)" represent possible alternatives to the recommended options.
(PDF)

**S9 Fig. Sharpness response as a function of lens position.** The graph measures the size of the JPEG stream that is decoded on the image to evaluate sharpness computationally.
(PDF)

**S1 Video. Incubator-contained timelapse video.** Multi-day time-lapse series of HeLa cells in an incubator using the ESPressoscope.
(MPG)

**S2 Video. Timelapse under water.** Multi-day time-lapse series where we placed the anglerfish in the creek "Leutra" in Jena to observe biofilm growth.
(MPG)

**S3 Video. Microfluidic imaging.** Real-time video of the web interface, where we control the flowrate of the 3D printed perestaltic pump that runs the ESPlanktoscope.
(MP4)

## Acknowledgments

We thank Stephan Saalfeld from the HHMI Janelia Research Farm and the AQLM Course at the MBL in Woods Hole MA, USA for conducting a workshop on building a series of ESPressoscopes.

## Author Contributions

**Conceptualization:** Ethan Li, Vittorio Saggiomo, Wei Ouyang, Manu Prakash, Benedict Diederich.

**Data curation:** Vittorio Saggiomo, Benedict Diederich.

**Formal analysis:** Vittorio Saggiomo, Wei Ouyang, Benedict Diederich.

**Funding acquisition:** Benedict Diederich.

**Investigation:** Ethan Li, Vittorio Saggiomo, Benedict Diederich.

**Methodology:** Ethan Li, Vittorio Saggiomo, Manu Prakash, Benedict Diederich.

**Project administration:** Benedict Diederich.

**Resources:** Vittorio Saggiomo, Benedict Diederich.

**Software:** Ethan Li, Wei Ouyang, Benedict Diederich.

**Supervision:** Benedict Diederich.

**Validation:** Ethan Li, Vittorio Saggiomo, Benedict Diederich.

**Visualization:** Vittorio Saggiomo, Benedict Diederich.

**Writing – original draft:** Ethan Li, Vittorio Saggiomo, Benedict Diederich.

**Writing – review & editing:** Ethan Li, Vittorio Saggiomo, Manu Prakash, Benedict Diederich.

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
