## [Decision Letter · Decision Letter 0]

25 Apr 2024

PONE-D-24-07702ESPressoscope: a small and powerful platform for in situ microscopyPLOS ONE

Dear Dr. Diederich,

Thank you for submitting your manuscript to PLOS ONE. After careful consideration, we feel that it has merit but does not fully meet PLOS ONE’s publication criteria as it currently stands. Therefore, we invite you to submit a revised version of the manuscript that addresses the points raised during the review process.

Attached are two documents with comments from the reviewers.

We look forward to receiving your revised manuscript.

Kind regards,

Manob Jyoti Saikia, Ph.D.

Academic Editor

PLOS ONE

Journal Requirements:

4. Please note that funding information should not appear in any section or other areas of your manuscript. We will only publish funding information present in the Funding Statement section of the online submission form. Please remove any funding-related text from the manuscript.

   "This research was supported by a Grant from the German-Israeli Foundation for Scientific Research and Development (GIF, Grant number  G-1566-413.13/2023)."

6. We note that Figure 7 (c) in your submission contain copyrighted images. All PLOS content is published under the Creative Commons Attribution License (CC BY 4.0), which means that the manuscript, images, and Supporting Information files will be freely available online, and any third party is permitted to access, download, copy, distribute, and use these materials in any way, even commercially, with proper attribution. For more information, see our copyright guidelines: http://journals.plos.org/plosone/s/licenses-and-copyright.

a. You may seek permission from the original copyright holder of Figure 7 (c) to publish the content specifically under the CC BY 4.0 license. 

Reviewers' comments:

Reviewer's Responses to Questions

**Comments to the Author**

1. Is the manuscript technically sound, and do the data support the conclusions?

Reviewer #1: Yes

Reviewer #2: Yes

2. Has the statistical analysis been performed appropriately and rigorously? 

Reviewer #1: N/A

Reviewer #2: Yes

3. Have the authors made all data underlying the findings in their manuscript fully available?

Reviewer #1: Yes

Reviewer #2: Yes

4. Is the manuscript presented in an intelligible fashion and written in standard English?

Reviewer #1: Yes

Reviewer #2: Yes

5. Review Comments to the Author

Reviewer #1: 

The article addresses the disadvantages of microscopy and proposes ESPressoscope, an alternative, inexpensive, and portable imaging system, and discusses microscopy solutions that enable new and more accessible experiments for biological and analytical applications. Please consider the comments reported in the comprehensive review report is attached.

Sincerely yours,

Reviewer

Reviewer #2: The manuscript proposes a novel and efficient method utilizing a simple microcontroller board, not only as a portable and field-ready digital microscope but also as a versatile platform adaptable to various imaging applications. I believe the manuscript is well-written and merits publication. However, there are two minor revisions needed. Firstly, the quality of figures is low and does not meet standard requirements. Secondly, while the "Conclusions and Outlook" section discusses the limitations and drawbacks of the method, adding experiments and analyzing reasons for the proposed method's poor performance in some cases can enhance the manuscript's value. This addition is highly recommended.

6. PLOS authors have the option to publish the peer review history of their article (what does this mean?). If published, this will include your full peer review and any attached files.

Reviewer #1: No

Reviewer #2: No

---

## [Author Response · Author response to Decision Letter 0]

20 May 2024

Reviewers' comments:

Reviewer's Responses to Questions

Comments to the Author

1. Is the manuscript technically sound, and do the data support the conclusions?

Reviewer #1: Yes

Reviewer #2: Yes

2. Has the statistical analysis been performed appropriately and rigorously?

Reviewer #1: N/A

Reviewer #2: Yes

3. Have the authors made all data underlying the findings in their manuscript fully available?

The PLOS Data policy requires authors to make all data underlying the findings described in their manuscript fully available without restriction, with rare exception (please refer to the Data Availability Statement in the manuscript PDF file). The data should be provided as part of the manuscript or its supporting information, or deposited to a public repository. For example, in addition to summary statistics, the data points behind means, medians and variance measures should be available. If there are restrictions on publicly sharing

https://zenodo.org/records/11179311 data—e.g. participant privacy or use of data from a third party—those must be specified.

Reviewer #1: Yes

Reviewer #2: Yes

4. Is the manuscript presented in an intelligible fashion and written in standard English?

Reviewer #1: Yes

Reviewer #2: Yes

5. Review Comments to the Author

Reviewer #1: 

The article addresses the disadvantages of microscopy and proposes ESPressoscope, an alternative, inexpensive, and portable imaging system, and discusses microscopy solutions that enable new and more accessible experiments for biological and analytical applications. Please consider the comments reported in the comprehensive review report is attached.

Sincerely yours,

Reviewer

6. PLOS authors have the option to publish the peer review history of their article (what does this mean?). If published, this will include your full peer review and any attached files.

Do you want your identity to be public for this peer review? For information about this choice, including consent withdrawal, please see our Privacy Policy.

Reviewer #1: No

Reviewer #2: No

Reviewer #1 

Dear Editor, Dear PLOS ONE Editorial Office, I would like to thank you for your kind invitation to review the manuscript ID# PONE-D24-07702 and entitled “ESPressoscope: a small and powerful platform for in situ microscopy”. Please find the review report below. The article addresses the disadvantages of microscopy and proposes ESPressoscope, an alternative, inexpensive, and portable imaging system, and discusses microscopy solutions that enable new and more accessible experiments for biological and analytical applications. Please consider the comments below to improve the paper further: 

From my perspective, the abstract section is well handled and conveys clear content about the article to the target readership. 

We thank the reviewer for the kind words and appreciate that.

The integration of firmware, user interface, and a browser-based graphical user interface (GUI) represents substantial contributions to both the paper and the broader scientific research within this field. These components not only enhance the functionality and usability of the proposed system but also pave the way for innovative advancements in the intersection of technology and scientific inquiry. 

Thank you!

3) Introduction has provided some background research and highlighted their advantages and disadvantages. Regarding the Introduction section, it is recommended to include references for the content discussed in lines 42-48 to ensure transparency and provide readers with the necessary background information. This would enhance the credibility and academic rigor of the manuscript. “Other microscopes have been developed with the goal of making specific research techniques more accessible at high performance and low cost. One example is the 3Dprinted and open-source OpenFlexure microscope, which is backed by a large user community (estimated 1k users) and is one of the first medically-certified microscopes used to identify malaria in blood smears . Another example is the Planktoscope,….” 

Outside of traditional research facilities, the same advances in consumer technologies have enabled affordable, user-friendly microscopes that can easily be customized \\cite{Bowman2023, Diederich2022} to meet specific experimental needs \\cite{Hohlbein2022}, and also to make microscopy accessible to a wider range of researchers and students. Many of these microscopes have been designed to promote STEAM education and research in previously underserved communities \\cite{Zakoth2019, Cybulski2014}. An excellent example is the 2 dollar Foldscope \\cite{Cybulski2014} and its associated global community for citizen scientists to share what they see in the microcosmos. Other microscopes have been developed with the goal of making specific research techniques more accessible at high performance and low cost. One example is the 3D-printed and open-source OpenFlexure microscope, which is backed by a large user community (estimated 1k users based on number from the official forum \\url{http://openflexure.discourse.group}, YouTube videos and manuscripts based on the OpenFlexure microscope), and is attempting be one of the first open-source medically-certified microscopes used to identify malaria in blood smears (\\url{https://openflexure.org/about/medical-devices}). Another example is the Planktoscope \\cite{Pollina2022}, a low-cost autonomous stop-flow imaging microscope controlled by a Raspberry Pi to quantify the local biodiversity of plankton using a simple yet powerful fluidic system and microscopy components. This is backed by a very active user base inside a Slack channel (\\url{https://www.planktoscope.org/join}) and an open-sourced commercialization of the device (\\url{www.fairscope.org}). These examples show that global impact is achievable for high-performance devices which are made open and affordable, and which develop large user communities . This strategy aims to increase the use of microscopy in life sciences and beyond by providing accessible tools for education, medical diagnosis, and research.

4) Furthermore, it is suggested that the Introduction section be expanded to encompass a broader literature review. Currently, the review predominantly focuses on a limited number of handmade microscopes, while the title suggests a wider scope with the phrase '...platform for in situ microscopy'. In the existing literature, there have been documented instances of miniature imagers constructed from webcams or low-cost sensors, offering convenient assembly and cost-effective in situ imaging of biological structures. Therefore, it is advisable for the authors to consider evaluating these systems to enhance the comprehensiveness of their study. It is recommended that the following articles be assessed for their relevance to similar studies, as an example: 

Tseng, D., Mudanyali, O., Oztoprak, C., Isikman, S. O., Sencan, I., Yaglidere, O., & Ozcan, A. (2010). Lensfree microscopy on a cellphone. Lab on a Chip, 10(14), 1787-1792. 

Polat, A., Hassan, S., Yildirim, I., Oliver, L. E., Mostafaei, M., Kumar, S., ... & Zhang, Y. S. (2019). A miniaturized optical tomography platform for volumetric imaging of engineered living systems. Lab on a Chip, 19(4), 550-561. 

Greenbaum, A., Luo, W., Su, T. W., Göröcs, Z., Xue, L., Isikman, S. O., ... & Ozcan, A. (2012). Imaging without lenses: achievements and remaining challenges of wide-field on-chip microscopy. Nature methods, 9(9), 889-895. 

Kim, S. B., Bae, H., Cha, J. M., Moon, S. J., Dokmeci, M. R., Cropek, D. M., & Khademhosseini, A. (2011). A cell-based biosensor for real-time detection of cardiotoxicity using lensfree imaging. Lab on a Chip, 11(10), 1801-1807. Zhu, H., Yaglidere, O., Su, T. W., Tseng, D., & Ozcan, A. (2011). 

Cost-effective and compact wide-field fluorescent imaging on a cell-phone. Lab on a Chip, 11(2), 315-322. Polat, A., & Göktürk, D. (2022). An alternative approach to tracing the volumic proliferation development of an entire tumor spheroid in 3D through a mini-Opto tomography platform. Micron, 152, 103173. 

We thank the reviewer for this great suggestion. We added this to the introduction:

A thorough review of handheld devices and imaging platforms for diagnostic, educational and professional use can be found in \\cite{C1LC20098D,C0LC00358A,Greenbaum2012} or in our review \\cite{WangHeintzmannDiederich}.

5) I found Figure 1 to be lacking in descriptiveness from the reader's perspective. It wasn't clear to me whether Matchboxscope, Anglerfish, ESPlanktoscope, ESPectrophotometer, and HoloESP in Figure 1 represent components of the ESPressoscope platform, or if the ESPressoscope platform transforms into a different model depending on their integration. Clarification on this aspect would enhance understanding. The caption is also recommended to be revised accordingly. 

Indeed, the figure makes it hard to understand which is the core component of any of the configurations and what is a standalone version. We have completely redesigned Figure 1 to clarify the relationship between ESPressoscope and Matchboxscope, Anglerfish, ESPlanktoscope, ESPectrophotometer, and HoloESP. We have also updated the caption for Figure 1 as follows:

The ESPressoscopThe ESPressoscope concept (left) consists of an ESP32 microcontroller development board with an embedded graphical user interface and an integrated camera which is combined with other modules which are selected depending on imaging configuration and which are combined through a layered structure. This paper demonstrates the concept with a prototype set of modules (center, refer to Fig. S1,) which we combined in various ways to achieve a variety of prototype optical configurations, such as a compact general-purpose microscope (Matchboxscope), an underwater microscope (Anglerfish), a flow-imaging microscope with an embedded fluidic device (ESPlanktoscope), a spectrophotometer (ESPectrophotometer), and a lensless holographic microscope (HoloESP).r development board with an integrated camera, 3D printed parts, and a few mechanical parts. The core unit of any imaging configuration represented by the ESP32 camera microcontroller, and a base which caries the electronics and can adapt additional components (Matchboxscope). It is possible to fabricate more complex imaging units such as underwater microscopes (Anglerfish), a microscope with an embedded fluidic device (ESPlanktoscope), a spectrophotometer (ESPectrophotometer), and a lensless holographic microscope (HoloESP) by adding auxiliary components such as light-sources, sample stages, etc. The optical path is visualized as a ray diagram, which also indicates the resulting magnification ($a$ represents the distance from object to the lens, $a'$ the distance between the lens and the camera, $d$ the overall distance between sensor and sample $d=-a+a'$)

6) In lines (133-135), there is some complexity regarding the variables used. It would be beneficial to clarify the meaning of 'd' and whether '≈ 4mm' also refers to 'a’'. Additionally, further explanation is needed on how the effective pixel size was calculated. An image on the sensor is focused and magnified if a < a0. However, to create a compact device 133 with the ESP32-CAM board’s d 4 mm sensor and f’ 4 mm focal length of the objective, we have 134 chosen a’ ≈ 18mm and ≈ 4mm, leading to a total magnification of 4 and an effective pixel size 135 of ≈ 0.6μm. 

That was indeed misleading. Therefore, we added Figure S1 that further explains the different distances mentioned in the equations. added a new figure in Supplementary S8 and explained the distances given by the formular stated above

7) Similarly, what is 0.23 and 1.2 µm in line 137?

We apologize for our fault. The math formatting did not show the correct elements. Thanks for clarification. We updated the text become:

Then the ESP32-CAM board’s f-number of ~#2.2 allows a numerical aperture of NA = 1/(2*2.2~0.23 and thus an optical resolution of dλ = 550nm=2*NA≈ 1.2μm.

8) In Fig 2 caption, there should be the explanation of a). Additionally, how the graph in Fig 2c created? I think it is not USF Chart Group of 6,7; it is 1D profile of USF chart… 

We apologise for the incorrect caption of Fig 2. We have updated the text to become:

(a) A Matchboxscope configuration featuring periscopic illumination and spring-based focusing mechanism. (b) This simple microscope can resolve features as small as 4-5µm inside the USAF chart (group 6,7) also demonstrated with (c) the lineplot along the vertical and horizontal direction (red, blue). Examples of micrographs obtained with the Matchboxscope: (d, e) A mosquito larvae found in a pond and (f) red blood cells.

9) If I am not mistaken, I could not find the supplementary Figures in the file. 

10) Is supplement Figure 2 Figure S2? Is Figure S1 supplement Figure 1? I recommend using Figure S1, Figure S2... by eliminating these confusions and following a standardization throughout the article. 

Thank you very much for the great suggestion, we changed this accordingly. 

11) There is no a) in Figure 5. The all captions should be reviewed and revised. 

Thank you very much for pointing to the issue, we have added the a) and revised all captions accordingly. 

12) The manuscript adeptly elucidates the conclusion, limitations, and avenues for future work. The conclusion effectively summarizes key findings while the discussion of limitations provides valuable insights into the study's scope and potential areas for improvement. Furthermore, the delineation of future work highlights the authors' forward-thinking approach, setting the stage for continued advancements in the field. 

We thank the reviewer for her or his appreciation of these passages. 

13) I would like to offer a humble suggestion to the authors to conduct a thorough review of the entire article, focusing on eliminating minor errors and revising any parts that may be difficult to understand. Additionally, I recommend revisiting the flow of the subject matter where necessary, reassessing the priorities in the sequence of Figure descriptions within the text to enhance coherence, and reviewing

---

## [Decision Letter · Decision Letter 1]

21 Jun 2024

ESPressoscope: a small and powerful approach for in situ microscopy

PONE-D-24-07702R1

Dear Dr. Diederich,

We’re pleased to inform you that your manuscript has been judged scientifically suitable for publication and will be formally accepted for publication once it meets all outstanding technical requirements.

Kind regards,

Manob Saikia, Ph.D.

Academic Editor

PLOS ONE

---

## [Editor Report · Acceptance letter]

2 Jul 2024

PONE-D-24-07702R1 

PLOS ONE

Dear Dr. Diederich, 

I'm pleased to inform you that your manuscript has been deemed suitable for publication in PLOS ONE. Congratulations! Your manuscript is now being handed over to our production team.

Kind regards, 

on behalf of

Dr. Manob Jyoti Saikia 

Academic Editor

PLOS ONE